# Mutation-induced infections of phage-plasmids

Xiaoyu Shan[1], Rachel E. Szabo[1,2] & Otto X. Cordero [1] ✉

Phage-plasmids are extra-chromosomal elements that act both as plasmids and as phages, whose eco-evolutionary dynamics remain poorly constrained. Here, we show that segregational drift and loss-of-function mutations play key roles in the infection dynamics of a cosmopolitan phage-plasmid, allowing it to create continuous productive infections in a population of marine *Roseobacter*. Recurrent loss-of-function mutations in the phage repressor that controls prophage induction leads to constitutively lytic phage-plasmids that spread rapidly throughout the population. The entire phage-plasmid genome is packaged into virions, which were horizontally transferred by re-infecting lysogenized cells, leading to an increase in phage-plasmid copy number and to heterozygosity in a phage repressor locus in re-infected cells. However, the uneven distribution of phage-plasmids after cell division (i.e., segregational drift) leads to the production of offspring carrying only the constitutively lytic phage-plasmid, thus restarting the lysis-reinfection-segregation life cycle. Mathematical models and experiments show that these dynamics lead to a continuous productive infection of the bacterial population, in which lytic and lysogenic phage-plasmids coexist. Furthermore, analyses of marine bacterial genome sequences indicate that the plasmid backbone here can carry different phages and disseminates trans-continentally. Our study highlights how the interplay between phage infection and plasmid genetics provides a unique eco-evolutionary strategy for phage-plasmids.

A key distinction among temperate phages is whether they integrate into the host chromosome (e.g., the well-known *Escherichia coli*'s phage lambda) or replicate as an extrachromosomal element. In this latter group are phage-plasmids, circular elements that appear to have evolved by the fusion of a plasmid and phage. Although a few examples such as phage P1[1] infecting *Escherichia coli* and phage VP882[2] infecting *Vibrio cholerae* have been extensively-studied, it is only very recently that we have become aware of the prevalence and relevance of these hybrid elements[3]. Recent surveys have found that phage-plasmids are abundant[3,4] and carry a large diversity of clinically relevant antibiotic resistant genes across bacteria[5]. Despite their significance, however, most of these elements have not been experimentally characterized

and their ecology and evolution as not only phages but also plasmids remain poorly understood.

As most other plasmids, phage-plasmids can also be found in multiple copies per cell (polyploidy). This fact has surprising implications for the population genetics of these elements and their dynamics of infection. Polyploidy makes it possible for cells to be heterozygous at any phage-plasmid encoded locus[6–8], including key genes such as the transcriptional repressor that maintains the phage in its lysogenic state. This intra-cell genetic variation can have a significant impact on phage-plasmid dynamics. If the prophage was chromosomally integrated, loss of function mutations in the phage repressor would be effectively suicide mutations, committing the phage to a lytic phage. However, in

[1]Department of Civil and Environmental Engineering, Massachusetts Institute of Technology, Cambridge, MA, USA. [2]Microbiology Graduate Program, Massachusetts Institute of Technology, Cambridge, MA, USA. ✉e-mail: ottox@mit.edu

theory, such defective allele variants could be recessive in a polyploid phage-plasmid. Polyploidy also implies that the intergenerational dynamics of phage-plasmids are affected by segregational drift—i.e., the random assortment of plasmid copies among daughter cells after cell division. Segregational drift can lead to large fluctuations in the degree of heterozygosity (including the production of homozygous offspring) in subsequent generations[9–11]. As shown below, the interplay between heterozygosity and segregational drift in phage-plasmids can lead to a type of eco-evolutionary dynamics unique for phage-plasmids.

We explore these dynamics focusing on a cosmopolitan type of phage-plasmid widespread among marine *Roseobacter*—an abundant copiotroph in the ocean[12]. Using a combination of experiments and mathematical models we show that the hybrid nature of phage-plasmids allows loss-of-function mutations in the phage repressor gene to be maintained in the population, leading to continuous productive infections. We show that phage-plasmid variants transmit rapidly throughout the population via horizontal transfer, increasing ploidy and producing heterozygous cells. This force is counterbalanced by segregational drift, which restores homozygosity. The combination of these forces leads to the continuous production of phages and the stable coexistence of infected and resistant cells. We continue to show that the phage-plasmids such as this are formed frequently in the environment via fusion of plasmid backbones and phages and widespread across disparate geographic regions, suggesting a successful life-style strategy for these parasitic elements.

## Results

### Recurrent productive infection of a phage-plasmid in *T. mobilis* after ~40 generations

*Tritonibacter mobilis* (previously known as *Ruegeria mobilis*) is a member of the *Roseobacter* clade[12], which collectively represents one of the most ubiquitous groups of marine heterotrophic bacteria[13]. *Tritonibacter mobilis* A3R06, carrying a temperate phage-plasmid, was isolated from an agarose particle inoculated with coastal seawater bacterial communities. Its genome has 4.65 million base pairs (Mbp), with a chromosome of 3.2 Mbp plus four (mega)plasmids of 1.2 Mbp, 0.1 Mbp, 78,000 base pairs (Kbp), and 42 (Kbp), respectively (Fig. S1). The 42 Kbp plasmid is also a circular phage with 51 predicted genes. These include genes encoding a phage head, tail, capsid and portal proteins, lysozyme, cell wall hydrolases as well as a C1-type phage repressor. The rest of the genes are involved in plasmid stability, replication and segregation, such as the *yoeB-yefM* toxin-antitoxin system, the *parAB* plasmid segregation system and P4-family plasmid primase (Fig. S2).

When growing *Tritonibacter mobilis* A3R06 under serial dilution cycles of approximately 6 generations per transfer in minimal media, we observed a reproducible, sharp decline in optical density (OD600) after approximately 40 generations (Fig. 1, Fig. S3). The decline in OD600 was due to the formation of cell clumps containing extracellular DNA (eDNA) (Fig. S5), consistent with the idea that cell lysis promoted clump formation[14]. By sequencing the time courses and analyzing differences in genome coverage, we confirmed the induction of the phage-plasmid in all of the 15 independent biological replicates after 30-50 generations (Fig. 1a, Table S1 and Fig. S4). To further validate the lysogeny-lysis switch of the phage-plasmid, we did transmission electron microscopic imaging of the 0.22 μm filtered supernatant from the clumpy bacterial culture, confirming the production of virion particles. The phage particle has *Siphoviride*-type morphology, with an isometric head of ~50 nm diameter and a long tail of ~180 nm length (Fig. 1b).

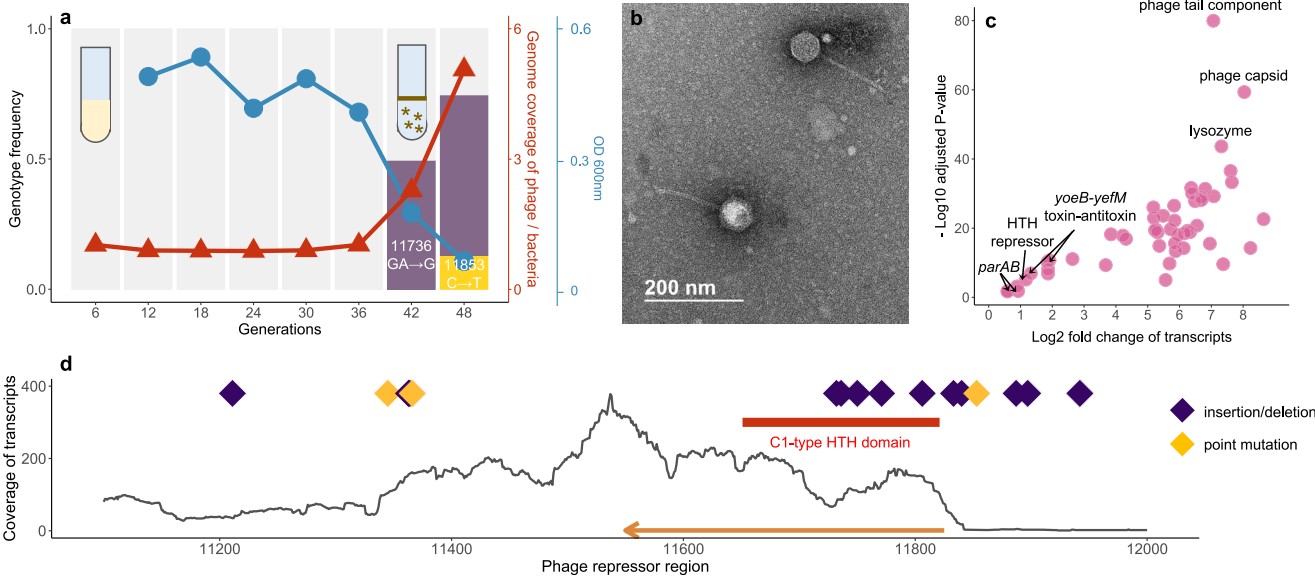

**Fig. 1 | Mutations in a phage repressor region recurrently drives productive infection of the phage-plasmid. a** Productive switch of a phage-plasmid in *Tritonibacter mobilis* A3R06 was observed after ~40 generations of serial-dilution growth (red). A deletion mutation (11736: GA→G, purple bar) rapidly increased to ~50% relative genotypic frequency within one dilution cycle, before the increase slowed down in the next dilution cycle. A second mutation (11853: C→T, yellow bar) was observed in the last dilution cycle. Planktonic bacterial culture became highly clumpy after the productive switch, as indicated by the sharp decrease in OD600 (blue). For eco-evolutionary trajectories for the other 8 populations temporally-tracked with genomic sequencing, see Fig. S4 and Table S1. **b** Transmission electron microscope image of the phage-plasmid particle. Imaging was performed for seven times with biological triplicates, all yielding similar results. **c** Differential expression of phage-plasmid genes before and after observing the mutation. Genes related to phage production were significantly upregulated after the productive switch, in particular the phage structural genes and the phage lysozyme gene. Expression of genes that are housekeeping for plasmid replication and stability were only increased because of copy-number increase of genes. P values are calculated based on Wald test and are adjusted by the Benjamini-Hochberg (BH) procedure. **d** All 21 mutations identified in 15 independent lines of populations were within a short ~1000 bp region encoding a C1-type phage repressor (orange arrow). Most of mutations are insertions or deletions (purple diamond). Details of these mutations are listed in Table S1. Source data for Figs. 1a, 1c and 1d are provided in the Source Data file.

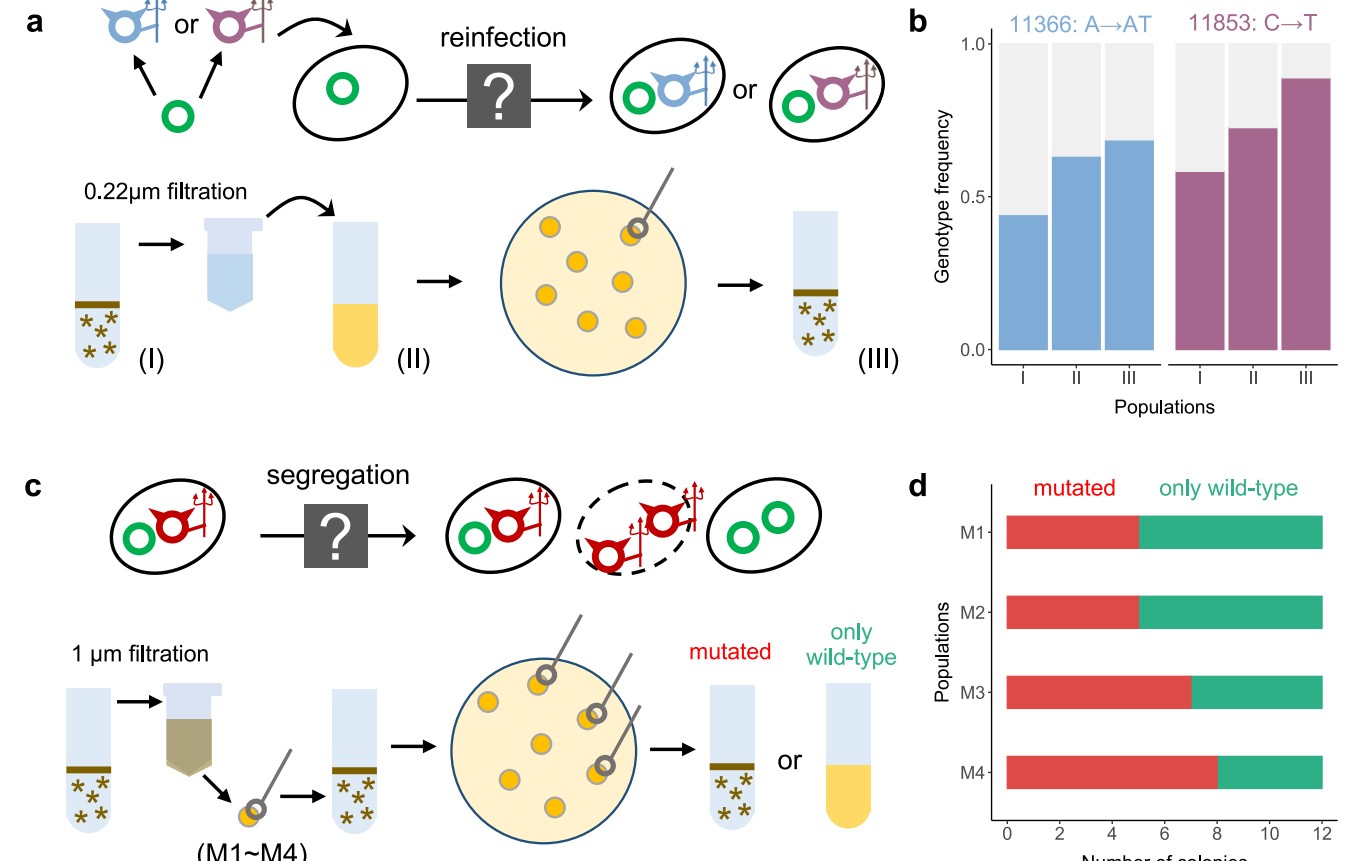

**Fig. 2 | Experimental confirmation of reinfection and segregational drift.**
**a** Schematic illustration of reinfection, for which we hypothesize that mutated phages are able to infect hosts lysogenized by wild-type phages. Wild-type and mutated phage-plasmids were showed as green circles and blue/purple/red color circles. See Methods for full experimental details. **b** Mutated phages with the same genotype were observed across (I) the initial source host population carrying the mutated phage, (II) the victim host population re-infected by the mutated phage and (III) the descendant of (II), supporting our hypothesis of reinfection. Experiments were performed in biological duplicates with two source host populations

carrying different genotypes of mutated phages (blue and purple). **c** Schematic illustration of segregational drift, for which we hypothesize that a host infected by mutated phages is able to generate offspring with only wild-type phages. See Methods for full experimental details. **d** Each of the 4 single-cell mother colonies (M1–M4) carrying a mixture of mutated phages and wild-type phages was able to generate descendants only carrying the wild-type phages, supporting our hypothesis of segregational drift. Source data of Fig. 2 are provided in the Source Data file.

## Productive infection of the phage-plasmid is driven by mutations in a phage repressor region

The observed 30-50 generations lag before the productive infection suggested that the lysogenic-lytic switch was less likely to be driven by metabolite accumulation or physiological signaling. Alternatively, we hypothesized that the prophage induction was driven by genotypic changes. To test this hypothesis, we did genomic sequencing for nine independent bacterial populations at the end of each dilution cycle. We found that only a short, ~1000 bp region in the phage genome encoding a C1-type phage repressor, consistently contained mutations across all the independent lines (Fig. 1d, Table S1). A large fraction (15/19, 78.9%) of the mutations were insertion/deletion mutations leading to frame-shift within either the helix-turn-helix DNA binding domain of the C1-type repressor or the putative upstream promoter region as inferred from the level of transcripts (Fig. 1d), suggesting that the mutations resulted in a loss of repressor function. Interestingly, a majority of these insertion/deletion mutations (9/15, 60.0%) were related to tandem repeats sequences in the genome (e.g., nucleotide position 11897: from GAAAAA to GAAAA, Table S1), which act as mutational hotspots due to replication slippage[15–17], suggesting that these mutations occurred faster than the background rate. Interestingly, several previous studies have found that mutations of tandem

repeats can be often reversed[18–20], which might enable an evolutionary switch between lysis and lysogeny.

In order to learn more about the consequences of the repressor mutations, we performed RNA-seq experiments for three independent populations, which allowed us to quantify the transcription of phage coding sequences before and after the mutation was observed. We found that the expression of genes related to a lytic phage lifestyle in the phage-plasmid were highly up-regulated after the observation of mutations in the repressor sequence, such as the phage capsid, phage tail and lysozyme (128–256 folds, Fig. 1c), showing these genes related to phage production were indeed de-repressed after the loss-of-function mutations. In contrast, those genes related to plasmid replication and stability such as the *yoeB-yefM* toxin-antitoxin system and the *parAB* plasmid segregation system were only increased similarly with the copy-number increase of the phage-plasmid (2–4 folds, Fig. 1c). These genes were actively expressed even before the mutation was observed (Figure S6, Supplementary Data 1), suggesting that they were functioning for a lysogenic lifestyle.

DNA sequencing across different timepoints during dilution cycles showed that, after repressor mutations were detected, their frequency in the population increased at an extremely rapid rate. e.g., jumping to ~50% within one dilution cycle of six generations in a

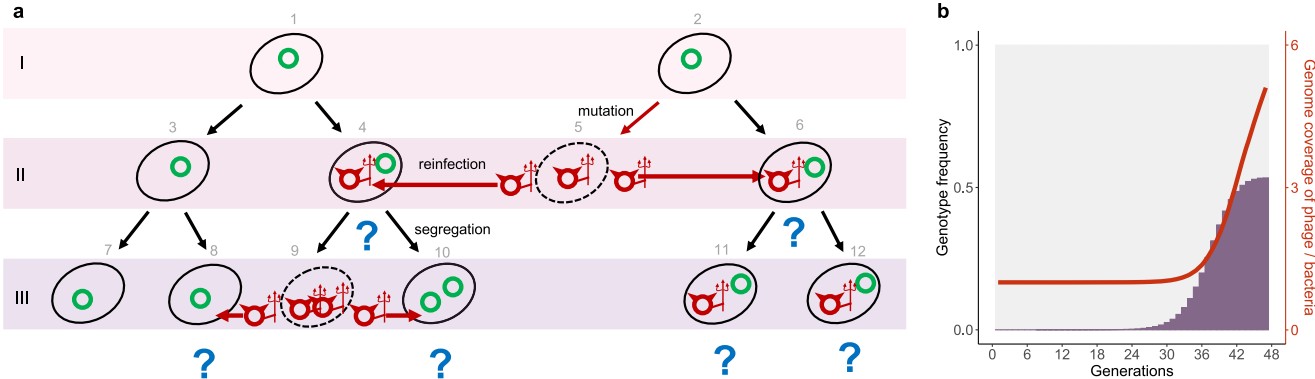

**Fig. 3 | A minimal model for phage-plasmid hybrid reproduces the observed eco-evolutionary dynamics. a** Schematic illustration of the model simulation. A question mark is drawn under a host cell when it carries more than one copies of phage-plasmids with different genotypes (a heterozygote). In those cases, the phage-plasmid genotype in descendants becomes stochastic due to segregational drift (e.g., cells 9, 10, 11, and 12). **b** With reinfection and segregational drift as the only two components, the simulated eco-evolutionary dynamics well matches the observed patterns in the experiment. For the experimentally observed eco-evolutionary dynamics, see Fig. 1a and Fig. S4. See Methods for full details of the model simulation. Source data of Fig. 3 are provided in the Source Data file.

population of ~10^8 cells. Despite this rapid increase, the mutant genotype never reached fixation, stabilizing at around 60% (Fig. 1a and Fig. S4). This pattern of evolutionary dynamics was intriguing in two respects. First, if transmission was only vertical, the drastic increase in the mutant genotype frequency would imply unrealistically high relative fitness coefficients ($s \sim 100$). Therefore, the evolutionary dynamics can only be explained by the infection spreading horizontally throughout the population. This presents an apparent conundrum, as we expected a host population lysogenized with the wild-type phage-plasmid to be immune to the same type of phage[21,22]. Second, if the mutated phage genotype was able to spread so rapidly throughout the population, why did not it reach fixation?

**Reinfection and segregational drift together explain the observed evolutionary dynamics**

We hypothesized that the observed evolutionary dynamics could be explained by the unique population genetic features of a phage-plasmid hybrid. If the mutated phage was able to spread through the population via virion production and reinfection, then the infected cells would likely contain multiple plasmid copies and be heterozygous at the repressor locus (e.g., one copy of lysogenic wild-type phage plasmid and one copy of mutated phage plasmid). In that case, segregational drift during the stochastic partitioning of plasmids between daughter cells should impact the subsequent bacterial and phage population dynamics. Indeed, recent studies have shown that the evolution of multiple-copy plasmids is affected by segregational drift[6,9,10], akin to the case of mitochondria in eukaryotes[11], resulting in variation in intracellular frequencies of plasmid-encoded alleles between mother cells and daughter cells. In the simplest scenario where plasmids were randomly distributed into daughter cells with equal opportunity, while the copy number of plasmids per cell remained constant, cell division could result in the maintenance of repressor heterozygosity at the single cell level, or the production of two homozygote cells, one carrying only wild-type and one carrying only mutated phage (Fig. S7). In the latter case, the daughter cell with only mutated phages would be lysed, releasing more mutated phage particles and continuing the spread of the phage.

Further experiments confirmed that the lytic phage-plasmid re-infected cells lysogenized with wild-type (Fig. 2a, b). To show this, we spiked cell-free supernatant containing the mutant phage-plasmid into a culture of the host carrying only the wild-type variant. After an overnight incubation we observed the appearance of clumps, identical to those that appear spontaneously after 30–50 generations (Methods). Genome sequencing of clones streaked out of this culture showed that they carried the full mutant phage-

plasmid, whose genotype were identical to the one of present in the cell-free supernatant used in the re-infection experiment, and that they were heterozygous at the repressor locus (Fig. 2b). In contrast, when we repeated the same experiment but with the cell-free supernatant 0.02 µm filtered to remove the phage particles, the victim host population remained planktonic, indicating no re-infection.

As a result of re-infection, we observed an increase in phage-plasmid copy number, opening the possibility for segregational drift to impact its evolutionary dynamics (Fig. 2c, d). Starting with a single colony carrying both the wild-type and mutant phage-plasmids, we questioned whether it was able to generate homozygous descendants, carrying only wild-type phages (Methods). As expected, we found that a heterozygous mother host cell was able to produce offspring that are free of the mutant phage. Genome sequencing of four post-segregation descendant populations confirmed that they only contained the wild-type phage-plasmid and its average copy number was significantly increased (Fig. S8, Kruskal-Wallis test $P = 0.02$) as a consequence of segregational drift (Fig. 2c).

A higher dosage of wild-type repressor gene copies should in principle provide a stronger buffer against phage-plasmid induction, at least in part because the probability of generating zero wild-type repressor after segregational drift would be lower. To test this, we restarted the serial dilution cycles with the post-segregation populations carrying a higher copy number of wild-type phages. Indeed, we found that it took at least 66 generations to observe phage-plasmid induction (Fig. S9), which was significantly longer than the 30–50 generations observed for wild-type populations with single-copy repressor gene.

With reinfection and segregational drift as the only two basic components, we found that a minimal probabilistic model was sufficient to reproduce the observed evolutionary dynamics of the phage-plasmid mutations (Fig. 3 and Methods for full details of simulation). We started with a population of one million host cells each carrying a single copy of wild-type phage-plasmid and doubling six times per serial passage exactly the same as in the experimental condition (Method). A loss-of-mutation happened in a random phage-plasmid, leading to the lysis of host cell and release of mutated phage-plasmid particles. Some of the released phage-plasmids managed to reinfect another randomly-encountered host based on an efficiency of re-infection ($R$), which is reminiscent of the production efficiency ($R_0$) in epidemiology (Methods). The re-infected hosts carrying heterozygous repressor loci underwent segregational drift, after which descendants carrying only mutated phage-plasmids were lysed and re-entered

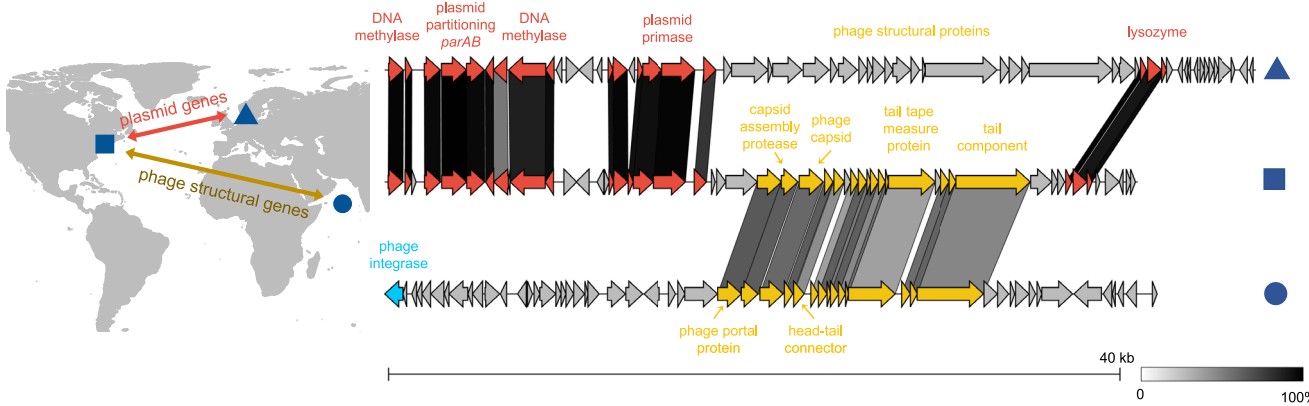

**Fig. 4 | The same plasmid backbone carrying different phages genes disseminate vast geographic distance.** *Tritonibacter mobilis* M41-2.2, isolated in Denmark, contains a phage-plasmid (triangle) whose plasmid-related genes are homologous and syntenic to those of *Tritonibacter mobilis* A3R06 phage-plasmid (square). However, their phage structural genes are very different from each other. Phage structural genes of *Tritonibacter mobilis* A3R06 phage-plasmid is both homologous and syntenic to that of a chromosome-integrated phage found in *Roseobacter* sp. SK209-2-6 (circle), which was isolated from deep water column in the Arabian Sea. Background map is from R package ggmap. Source data of Fig. 4 are provided in the Source Data file.

the infection cycle. With these ingredients, our simulation displayed a rapid spread of the mutated phage infection before quickly saturating, which is consistent with experimental observations (Fig. 3).

## Dissemination of phage-plasmids across geographic and phylogenetic distances

Using comparative genomics, we found that the plasmid backbone of the *Tritonibacter mobilis* A3R06 phage-plasmid carries different phage head and tail components across different continents (Fig. 4). To better understand the ecological distribution of phage-plasmids, we leveraged 1849 genomes available in the RefSeq database from the order of *Rhodobacterales*, to which the *Tritonibacter mobilis* was affiliated. With these genomes, we searched for homologs of the phage-related genes or plasmid-related genes in our *Tritonibacter mobilis* phage-plasmid (isolated in Massachusetts, USA). Strikingly, we found clusters of nearly identical (~100%) homologs of plasmid-related genes, such as *parAB* segregation system and P4-family plasmid primase, in another phage-plasmid of another *Tritonibacter mobilis* strain isolated in marine aquaculture in Denmark[12] (Fig. 4). These gene clusters were also found in perfect synteny, which strongly indicated a recombination event. However, the phage structural genes (e.g., phage head and tail) of these two phages were very different, both in terms of homology and synteny. The structural genes of our *Tritonibacter mobilis* phage-plasmid was both homologous and syntenic to those found in another phage integrated in the genome of a *Roseobacter* strain, which was isolated from 2500 m deep water in the Arabian Sea[23] (Fig. 4). Taking together, our results showed that the evolution of the plasmid-related genes and the phage-related genes for phage-plasmids could be decoupled. Different phages could become the genetic cargo of the same plasmid, which was able to transmit across continents carrying their phage components. This was consistent with recent findings showing that the core plasmid backbone could be recombined with different cargo genes in marine or human gut microbiome[24,25], with our findings suggesting that this could be exploited by phages to disseminate across large geographic distances. Additionally, we identified a second example of nearly identical plasmid-related genes but very different phage-related genes between two other phage-plasmids[26,27] (Fig. S11). These two phages were found in two strains of different though related bacterial species (average nucleotide identity ~92%), suggesting that plasmid backbones were also able to transmit across phylogenetic distances. All those phage-plasmids resemble *Tritonibacter mobilis* A3R06's phage-plasmid: a 40K−50K genome size with the presence of independent replication

systems (such as *ParABS* and *RepABC*) but absence of genes known for effective reinfection blockage (such as *SieA* of *E.coli* phage P1[22]), suggesting those phages can be also subject to mutation-driven induction in natural environment.

A bioinformatic search across publicly available metagenomes identified significant hits ($e < 10^{-15}$) of the *Tritonibacter mobilis* A3R06 phage-plasmid repressor in several natural microbial communities across the globe (Fig. S12). Hits were found mainly in nutrient rich marine environment, such as an shrimp aquaculture in the eastern coast of China[28], particle-attached Mediterranean water column[29], as well as an intense algal bloom in California[30]. Further studies are required to get more complete sequence information of phage-plasmids and to capture and characterize their eco-evolutionary dynamics in natural environments.

## Discussion

In this study, we found that mutation and segregational drift controlled the dynamics of transmission of a cosmopolitan phage-plasmid. First, we showed that a spontaneously mutated phage-plasmid was able to re-infect a host lysogenized with a wild-type phage, which prevented the mutant phage-plasmid from turning lytic. The occurrence of phage re-infection indicates that superinfection exclusion is not always effective, which is consistent with experimental observations for some other phages[31,32] and models suggesting a short-term evolutionary benefit of allowing super-infection[33,34]. Second, we showed that segregational drift diversified the phenotypic outcomes of daughter cells, with cells carrying only mutant phage-plasmids lysing and releasing virions. This highlights the impact that multi-copy elements like plasmids can have on the evolutionary dynamics of bacteria. Taken together, these observations reflect a mixture of both phage and plasmid properties: the phage facet enables rapid horizontal proliferation through virion production while the plasmid facet enables heterozygosity and segregational drift. Consequently, the phage-plasmids proliferated rapidly through iterative reinfection and lysis of a stochastically selected proportion of host descendants, leading to the co-existence of mutant phages and wild-type phage-plasmids. This strategy also reflects how genes and alleles can use viruses to rapidly propagate through a population without driving it to a sudden collapse.

The mutation-driven switch from lysogeny to lysis we observed in this study is distinct from the traditional model of prophage induction, in which lysis is triggered by a regulatory response to stress, chemical signals, etc. One important consequence of this difference is that the rate of mutation-driven induction becomes proportional to the rate of

mutation accumulation in a population. Therefore, a rapidly growing host population with large population sizes can develop continuous productive infections, as observed in this study. This feature can be relevant for "opportunitrophs" like members of the *Roseobacter* clade[35]. Members of this clade are known to frequently switch between two types of ecological life-styles, i.e., a survival mode in low-nutrient regions, and rapid growth mode on transient nutrient hotspots such as the phycosphere of marine algae[36]. Thus, we hypothesize that the mutational switch is a phage-plasmid adaptation to rapidly propagate through those fast-growing population, without driving them to a collapse.

We found that phage plasmids such as the one here described evolve by the rapid mixing and matching of plasmid backbones and prophages. This is evident in the fact that the phage region of the phage-plasmid was homologous to a phage found in the Arabian Sea, while its plasmid backbone was homologous to another phage-plasmid found in Denmark. Considering that the element that is the focus of this paper was isolated from the coast of Massachusetts, our findings suggest that phage plasmids might be evolutionary chimeras that combine elements with disparate evolutionary histories and disseminate across vast geographic distances. The wide distribution of these elements in natural environment and their ability to maintain continuous productive infections with rapid transmission of new genetic variants, suggest that these elements may be major vectors of horizontal gene transfer. Further work is needed to better-understand the ecological relevance of this hybrid elements and their potential of mutation-driven induction to trigger continuous productive infections in natural and synthetic systems.

## Methods

### Media
The minimal marine media, MBL media, was used for serial dilution growth of Tritonibacter mobilis A3R06. It contained 10 mM $NH_4Cl$, 10 mM $Na_2HPO_4$, 1 mM $Na_2SO_4$, 50 mM HEPES buffer (pH 8.2), NaCl (20 g/liter), $MgCl_2*6H_2O$ (3 g/l), $CaCl_2*2H_2O$ (0.15 g/l), and KCl (0.5 g/l). Glucose was added as the only carbon source at a concentration of 27 mM. Trace metals and vitamins were added by 1:1000 of the following stock solution. Trace metals stock solution included $FeSO_4*7H_2O$ (2100 mg/l), $H_3BO_3$ (30 mg/l), $MnCl_2*4H_2O$ (100 mg/l), $CoCl_2*6H_2O$ (190 mg/l), $NiCl_2*6H_2O$ (24 mg/liter), $CuCl_2*2H_2O$ (2 mg/l), $ZnSO_4*7H_2O$ (144 mg/l), $Na_2MoO_4*2H_2O$ (36 mg/l), $NaVO_3$ (25 mg/l), $NaWO_4*2H_2O$ (25 mg/l), and $Na_2SeO_3*5H_2O$ (6 mg/l). Vitamins, which were dissolved in 10 mM MOPS (pH 7.2), contained riboflavin (100 mg/l), D-biotin (30 mg/l), thiamine hydrochloride (100 mg/l), L-ascorbic acid (100 mg/l), Ca-D-pantothenate (100 mg/liter), folate (100 mg/l), nicotinate (100 mg/l), 4-aminobenzoic acid (100 mg/l), pyridoxine HCl (100 mg/l), lipoic acid (100 mg/l), NAD (100 mg/l), thiamine pyrophosphate (100 mg/l), and cyanocobalamin (10 mg/l). Marine Broth 2216, a rich media commonly used for growing marine bacterial strains, was purchased from the Fisher Scientific (BD 279110).

### Culture growth
*Tritonibacter mobilis* A3R06 was isolated from an agarose particle in coastal seawater from the Nahant Beach, Massachusetts, USA[37]. Single colonies of *Tritonibacter mobilis* A3R06 on Marine Broth agar plates were picked for enrichment in 2 mL liquid Marine Broth 2216 media for 6 h. After that, 50 μL of enriched culture was transferred into 4 mL MBL minimal media for pre-culture growth. Cells in the pre-culture was grown to mid-exponential phase before being diluted into 4 mL fresh MBL minimal media to an OD600 of roughly 0.01 to initiate the serial dilution cycles. Each cycle lasted for 24 h, corresponding to roughly six generations per cycle considering the doubling time of *Tritonibacter mobilis* A3R06 being ~4 h in MBL minimal media with glucose (Fig. S3). At the end of each cycle, cells were still within the exponential phase of growth (Fig. S3), except for those very late cycles where cell clump

formed following the productive switch of the phage-plasmid. All liquid culture growth was performed in Innova 42R incubator shaking at 220 rpm at 25 °C.

### DNA extraction, Illumina sequencing, and reads processing
Prior to DNA extraction, the cell culture samples were centrifuged at 8000 *g* for 60 s to remove the liquid. The cells were then resuspended into fresh MBL media by thoroughly pipetting for at least fifteen times. For each sample, the resuspension-centrifuge procedure was repeated for three times so as to wash away free virion particles outside of the cells. For Illumina sequencing, DNA was extracted with the Agencourt DNAdvance Genomic DNA Isolation Kit (Beckman Coulter). DNA concentration was quantified with Quant-iT PicoGreen dsDNA Assay kit (Invitrogen) on a Tecan plate reader. Short-read sequencing was performed on an Illumina NextSeq 2000 platform (2x151bp pair-ended). Library preparations and sequencing were performed at the Microbial Genome Sequencing Center (Pittsburgh, PA). Sequencing reads were trimmed to remove adaptors and low-quality bases (-m pe -q 20) with Skewer v0.2.2[38]. The remaining paired reads were checked for quality with FastQC v0.11.9.

### Closing the genome of *Tritonibacter mobilis* A3R06
Nanopore long-read sequencing was used to close the genome of Tritonibacter mobilis A3R06. DNA was extracted with a Qiagen DNeasy kit for higher DNA yield following the manufacturer's protocol. Long-read sequencing was performed on the Oxford Nanopore platform with a PCR-free ligation library preparation at the Microbial Genome Sequencing Center (Pittsburgh, PA). Closed genome of Tritonibacter mobilis A3R06 was assembled using Unicycler v0.4.9[39] by combining Illumina short reads and Nanopore long reads, resulting in one chromosome plus four circular plasmids. The assembly graph in gfa format was visualized by Bandage v0.8.1[40]. Coding sequences were predicted using prodigal v2.6.3[41] and functionally annotated with eggnog-mapper v2[42] (−go_evidence non-electronic−target_orthologs all−seed_ortholog_evalue 0.001−seed_ortholog_score 60). The phage genome map was visualized by SnapGene v6.0 (Insightful Science; available at snapgene.com).

### Read mapping and variant calling
The complete genome of Tritonibacter mobilis A3R06 was used as the reference genome. Quality-filtered pair-ended Illumina sequencing reads were mapped the reference genome using Minimap2 v2.17 with stringent settings (-ax sr)[43]. Genetic variants were identified from the aligned reads using BCFtools v1.13 with only variants with quality score ≥20 and a local read depth ≥20 were remained[44].

### RNA isolation and sequencing
RNA Protect Bacterial Reagent (Qiagen, Hilden, Germany) was added to the cell culture samples at a 2:1 volume ratio. RNA was isolated with a Qiagen RNeasy kit following the manufacturer's protocol except for the following changes:[45] cells were resuspended in 15 mg/mL lysozyme in TE buffer and incubated for 30 min at room temperature before adding buffer RLT. Mechanical disruption of samples using lysing matrix B (MPBio, Santa Ana, CA) were performed by shaking in a homogenizer (MPBio) for 10X 30 s intervals. Dry ice was added in the homogenizer to prevent overheating. Illumina Stranded RNA library preparation with RiboZero Plus rRNA depletion and pair-ended Illumina sequencing (2x51bp) were performed at the Microbial Genome Sequencing Center (Pittsburgh, PA).

### Transcriptomic analysis
Paired-end RNA reads were trimmed using Skewer v0.2.2 to remove sequencing adapters and low-quality reads (-m pe -q 20)[38]. The remaining paired reads were checked for quality with FastQC v0.11.9 and mapped to *Tritonibacter mobilis* A3R06 genome using Bowtie2

v2.2.6[46]. The generated SAM files were sorted by position using SAM-Tools v1.3.1[44]. Count table of transcripts were obtained by HTSeq v0.11.3[47] and differential gene expression was evaluated with DeSeq2 R package[48]. Normalized transcript abundance was generated from count tables by transcripts per kilobase million calculations.

## Fluorescence staining and light microscope

Live cells were stained by 5 μM SYTO9 which emits green fluorescence when it is bound to DNA. Dead cells were stained by 20 μM propidium iodide which emits red fluorescence when it is bound to DNA but was unable to permeate the cell membrane. Fluorescence was visualized using an ImageXpress high content microscope equipped with Meta-morph Software (Molecular devices, San Jose, CA), operating in widefield mode. Images were acquired in widefield mode at 40× with a Ph2 ELWD objective (0.6 NA, Nikon) and filter sets: Ex 482/35 nm, Em: 536/40 nm, dichroic 506 nm to detect SYTO9 and Ex 562/40 nm, Em 624/40 nm, dichroic 593 nm to detect propidium iodide. Images were collected with exposure times of 100 ms and processed with ImageJ v1.53[49].

## Transmission electron microscopic imaging

Transmission electron microscopic imaging was performed at Koch Institute's Robert A. Swanson (1969) Biotechnology Center Nano-technology Materials Core (Cambridge, MA). Samples were negatively stained with 2% uranyl acetate and were imaged on an JEOL 2100 FEG microscope. The microscope was operated at 200 kV and with a magnification in the ranges of 10,000–60,000 for assessing particle size and distribution. All images were recorded on a Gatan 2kx2k UltraScan CCD camera.

## Comparative genomics of Rhodobacterales phages

A total of 1849 genomes in the Family of *Rhodobacterales* were downloaded from NCBI RefSeq database on Jan 1st 2022. Coding sequences were annotated by eggnog-mapper v2 (–go_evidence non-electronic–target_orthologs all–seed_ortholog_evalue 0.001–seed_ortholog_score 60)[42]. Phage sequences were predicted with VIBRANT v.1.2.0[50]. MMseqs2[51] was used to search for homologs of *Tritonibacter mobilis* A3R06 phage genes, with high sensitivity parameters (-s 7.5 -c 0.8). The search output of MMseqs2 were sorted for the most significant hits as well as the highest number of hits, leading to the finding of *Tritonibacter mobilis* M41-2.2 phage and *Roseobacter* sp. SK209-2-6 phage. Alignment map was visualized with clinker clinker v0.0.23[52]. Map is visualized using the default world map in R package ggmap v3.0.0[53].

## Model simulation of eco-evolutionary dynamics

In order to simulate the evolutionary dynamics of the phage-plasmid, we developed a minimal model combining the population genetics of a plasmid as well as infection dynamics of a phage.

Our model in part resembles a classical Wright-Fisher model, which assumes non-overlapping generations in a discrete Markov process. However, we considered dynamic population size in our model, which incorporates cell doubling within a serial dilution cycle as well as cell lysis due to phage production. To start with, we have a population of $N_0$ host cells. When there is no phage-plasmid productive infection, all cells divide into two daughter cells thus the population size doubled every generation following $N_t = N_0 2^t$, which reaches $2^6 \times N$ at the end of each serial dilution cycle. Then a bottleneck indicated by the dilution factor $d$ was applied to the population so that $1/d$ cells were randomly sampled from the current population to enter the next serial dilution cycle. In our simulation, we use $N = 10^6$ at the beginning of each dilution cycle and dilution factor $d = 64$ as we did in experiment. Each host cell carries one copy of wild-type lysogenic phage-plasmid, replicating and segregating into two daughter cells as the host cell divides.

Loss of function mutations in the phage repressor gene result in this lysogenic phage becoming constitutively lytic. The host cell carrying the mutated phage-plasmid is then killed, releasing virion particles with the mutated phage genome to randomly infect other host cells. Previous studies have reported burst size of marine prokaryotic phages ranging from 4 to more than 100[54]. In our model, what matters in population genetics is the average number of released mutated phage-plasmids that successfully re-infect a host cell per host cell lysed. This parameter, termed as re-infection efficiency $R$, is similar to the parameter $R_0$ in epidemiology and should be lower than the empirical burst sizes[55], especially considering that the other host cells have been already lysogenized with a wild-type phage-plasmid larger than 40KB. We found that the saturating frequency of the mutated genotype was affected by $R$, for which $R = 5$ best fitted the experimentally observation (Fig. S10).

The host cells re-infected by the mutated phage-plasmids become heterozygote with more than one copies of phage-plasmids. Segregation of multiple copies of phage-plasmids with different genotypes can lead to genetic heterogeneity among daughter cells. In our minimal model, we assume a simplest scenario where phage-plasmids were randomly distributed into daughter cells with equal opportunity, while the copy number of plasmids per cell remained constant. For a cell host with $a$ copies of wild-type phage-plasmids and $b$ copies of mutated phage-plasmids, the segregation can be described using a Binomial distribution $B(2a + 2b, a + b)$. For instance, the probability of having $a_1$ copies of wild-type phage-plasmids in the first daughter cell follows

$$P(a_1) = C_{2a}^{a_1} C_{2b}^{a+b-a_1} / C_{2a+2b}^{a+b} \tag{1}$$

To simulate segregational drift at cell division, we performed Binomial sampling for all heterozygotic cell hosts containing more than one copies of phage-plasmids at each generation, generating daughter cells with stochastically different genotypes. Cells carrying at least one copy of wild-type phage-plasmid are prevented from lysis since the repressor gene is normally functioning. Host cells carrying only mutated phage-plasmids after segregational drift will be killed since the lytic genes on the phage-plasmids are no longer repressed. These lysed cells will be used to produce more virion particles to re-infect more host cells in the next cycle.

We implemented the simulation in R 4.1.0 and the code is available at https://github.com/Xiaoyu2425/PP (https://doi.org/10.5281/zenodo.7730856). The script simulates the life cycle of the phage-plasmid as detailed above, with the following parameters: initial population size ($N = 10^6$), re-infection efficiency ($R = 5$) and dilution factor ($D = 64$).

## Experimental confirmation of reinfection

To verify re-infection, mutated phage-plasmids were used to infect the host population carrying the wild-type phage-plasmid. Mutated phage-plasmids were separated from the source host population cells by filtering through a 0.22 μm pore size membrane. The supernatant was then spiked into a victim host population carrying only wild-type phage-plasmids growing in fresh MBL minimal media. The planktonic culture became highly clumpy after overnight growth, indicative of phage-plasmid induction. The infected population was then used to streak an agar plate for descendant single colonies. Colonies considered to harbor the mutated phage through re-infection, as indicated by the clump formation after re-growing in liquid media, were sequenced for phage-plasmid genotyping. The experiment was performed in biological duplicates with two source host populations carrying different mutated genotypes (11366: A→AT and 11853: C→T).

## Experimental confirmation of segregational drift

To verify segregational drift, host populations carrying both wild-type phages and mutated phages were tested for whether they were able to generate offspring with only wild-type phages. To ensure the host population really came from a heterozygote single cell rather than a clump of cells with mixed genotypes, we filtered the host population carrying mutated phages with 1 μm cell strainer (Pluriselect 437000103) to remove the multicellular clumpy aggregates. The filtered planktonic subpopulation was carefully examined under the microscope to ensure it contained planktonic cells clearly separated from each other. We then streaked this planktonic subpopulation on an agar plate for single colonies, of which 4 single colonies carrying both wild-type and mutated phages were picked as mother colonies. For each mother colony, liquid culture after overnight growth was then used to streak agar plates for daughter colonies, of which 12 daughter colonies were picked per mother colony. All the 48 daughter colonies were screened in liquid culture for whether they were planktonic, which is indicative of carrying only wild-type phage-plasmids, or clumpy, which is indicative of induction of mutated phage-plasmids. Further, we performed whole-genome sequencing of four daughter colonies that are planktonic in liquid culture, confirming that 1) they were indeed free of any mutated phage-plasmids and only contained wild-type phage-plasmids and 2) the copy number of phage-plasmid in their genomes are significantly increased.

## Phage susceptibility of other Rhodobacterales strains

The *yoeB-yefM* toxin-antitoxin system encoded by the phage-plasmid makes it difficult to cure the plasmid for the *Tritonibacter mobilis* A3R06 host. We therefore tried to test whether this phage-plasmid is able to infect any other bacterial host with a plaque assay, including *Tritonibacter mobilis* F1926 which is the model strain for the *Tritonibacter* genus[12] and other 28 *Rhodobacterales* isolates in the Cordero lab strain collection. However, none of those isolates were subject to infection. Recent studies suggested that specificity of phage infection may be related to the structure of bacterial capsule[32,56,57]. This may be the case for our phage-plasmid since *Tritonibacter mobilis* A3R06 harbors another 78Kb plasmid encoding a capsule, which was not found in the genome of other strains we tested for susceptibility.

## Reporting summary

Further information on research design is available in the Nature Portfolio Reporting Summary linked to this article.

## Data availability

The sequencing data generated from this study, including the closed genome of *Tritonibacter mobilis* A3R06, whole genome sequencing and transcriptomic sequences for experimental evolution populations have been deposited in NCBI under accession code PRJNA895449. Source data are provided with this paper.

## Code availability

R codes for mathematical simulation of phage-plasmids eco-evolutionary dynamics are available at github (https://github.com/Xiaoyu2425/PP) and Zenodo (https://doi.org/10.5281/zenodo.7730856).

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

## Acknowledgements
We thank Professors Lone Gram, Tami Lieberman and Terence Hwa for their help and suggestions to this work. We thank Gabriel Vercelli, Jie Yun, Fatima Hussain, Sammy Pontrelli, and all Cordero lab members for helpful discussions. We thank MIT Koch Institute's Robert A. Swanson (1969) Biotechnology Center for technical support, in particular Dr. Dongsoo Yun with the JEOL 2100 FEG microscope (RRID:SCR_018674). O.X.C. and X.S. were supported by the Simons Collaboration: Principles of Microbial Ecosystems, award number 542395. R.S. is supported by the NSF Graduate Research Fellowship under Grant 174530.

## Author contributions
X.S. and O.C. conceived the study. X.S. performed experiments. X.S. and R.E.S. performed data analysis. X.S., R.E.S. and O.C. wrote the manuscript.

## Competing interests
The authors declare no competing interests.
