## [Peer Review File · Nature Communications]

Mutation-induced infections of phage-plasmidsReviewer #1 (Remarks to the Author):

This very interesting manuscript describes a study on the evolution of lysis-lysogeny decision of phage-plasmids in an environmental bacteria. Phage-plasmids have recently spurred significant interest since they are more abundant than previously thought and have genetic traits of both phages (horizontal transmission) and plasmids (vertical transmission). Here it is shown that a phage-plasmid evolves to become more lytic and that it can super-infect hosts carrying the original phage-plasmid. This leads to heterozygosity in phage-plasmids which affect the fate of the cell. This is very interesting, because there are quite a few works on heterozygosity in plasmids but little or nothing in phages. The manuscript is clearly written and the figures illustrate well the results. Supplementary material is useful and almost complete (see below). Overall, I found this to be a very interesting and timely contribution.

-I was expecting some longer discussion on the observed evolution towards higher virulence in the context of the experience and how likely it is of being important in natural populations. I expect that the environment of the experiments selects for it, right? (phages have only lysogens to infect). How relevant is this outcome expected to be in nature?

-"we conclude that phage plasmids are evolutionary chimeras that combine elements with disparate evolutionary histories and disseminate across vast geographic distances " This general claim seems unwarranted given that the authors have analyzed only three elements. Furthermore, this analysis does not allow to orient the events, i.e. to determine which gave rise to what. This is important because plasmids and phages also recombine a lot. So, how these elements came about here it's not easy to infer. I think the observation of homology between phage-plasmid, phages and plasmids is interesting and worthy of note. The speculation that the phage-plasmid arose as a chimera has merit. But given the limited evidence, I don't think we can conclude from these results that this phage-plasmids (and even less these elements in general) originate from a chimera.

-L101. I was expecting some discussion around the fact that many of the mutations are slippage of tandem repeats. These mutations are often reversible, which could be relevant as a way of regulating the lysis-lysogeny switch (Moxon's, Current Biol, classical paper on contingency loci might be more relevant than the papers about mutations in autism). This mechanism might provide an interesting reversible mutational switch and sustain the points of discussion around lines 244.

-There are a number of recent works showing that super-infection exclusion is not as ubiquitous as previously thought, e.g. Bondy-Denomy, ISMEj, 16, Sousa, ISMEj, 20, etc. Also, there is theoretical work on the short and long-term advantages of exclusion in Hunter, PLoS CB, 22 and Wodarz, Evol Lett, 18. It would be interesting (and might increase the paper readership) to frame the discussion in this broader literature (even if it concerns other temperate phages).

-The section on the simulations should include a description of implementation and the code should be made available.

Minor points.

-L76. At this stage in the text it is not clear enough to the reader how productive infection was observed. In these studies one often delivers phages to observe productive infection. Not the case here, if I understood correctly. So, this should be more clearly explained from the onset. Maybe the issue about super-infection being tolerated in this system should be brought forward earlier in the text.

-Figure 1D should include other genetic elements in the annotation. It indicates one gene, but not the others, some of which might have mutations too (since some mutations are outside of the gene).

-Figure S6 would benefit if the most outstanding transcripts were labelled with gene names.

-The authors mention the availability of expression and genomic data upon publication. I think that one supplementary table is missing. It would have the genes of the phage-plasmid in lines, and in columns the information on their annotation (position, protein ID, and predicted function), information on those that have homologs in Fig4 and the levels of expression in the two conditions (fig S6). Otherwise it's difficult or impossible to access that information.

There are a few typos or issues with style:

-L32. Escherichia coli P11 infecting Escherichia coli

- L247. plasmids such the one here
- L356. used to search for search
- L376. switches to be lytic due to silence of the phage repressor

A few issues with references (see also above):

- L436. "Recent studies suggested that specificity of phage infection may be related to the structure of bacterial capsule " the references missing
- Some references have twice the year of publication: 6, 7, 8, etc
- Some preprints have been published or have different format from the rest: 5, 22, etc
- Some references have links in titles or other oddities: 11, 17, etc

Reviewer #2 (Remarks to the Author):

In this study, Shan, Szabo and Cordero investigate the evolutionary dynamics of a phage-plasmid in the marine bacterium *Tritonibacter mobilis*. The authors describe how the phage-plasmid became lytic after 35-40 generations of propagation of bacterial populations in the lab due to mutations in the phage repressor. However, after an extremely rapid increase in frequency, the mutant phage alleles did not reach fixation but stabilized at around 60% frequency in the population. These results suggested horizontal transfer of the derepressed phage-plasmid, producing heteroplasmy in reinfected cells carrying wild type phage-plasmids. The authors confirmed this possibility experimentally, and demonstrated how reinfection and segregational drift could explain the evolutionary dynamics observed. Finally, they also showed that similar phage-plasmids to the one described in their study (or phage-plasmid combinations carrying a very similar plasmid or phage section) are common and distributed around the world, suggesting a modular mode of evolution of phage-plasmid combinations, and the generalized relevance of the eco-evolutionary dynamics described in this study. In general, I think this is a very interesting work, providing new insights into phage-plasmid biology and their impact in bacterial evolution. I'm a big fan of the role of multicopy plasmids in bacterial evolution, and this study provides yet another fascinating example of the relevance of copy number of extrachromosomal genetic elements determining evolutionary dynamics in bacteria. In summary, I have enjoyed reviewing this paper a lot and I think it will represent an important contribution to the field. There is only one aspect of the paper that I don't fully understand (see comment below).

Major comment

I have to start by stating that I am an expert on plasmid biology, so some of the aspects of phage biology may be beyond my knowledge.

In the wild-type cells the copy number of phage-plasmid per chromosome is close to one. Once the mutations in the repressor appear, the phage-plasmid copy number (PPCN) goes up, which is probably mostly due to the active lytic cycle in some of the cells. However, it is also possible that in heterophage-plasmid cells (cells with both wild-type and mutated phage-plasmids) the PPCN can be increased due to the dilution of the wild-type repressor, which is only produced by the wild-type phage-plasmid, but has to bind both the wild-type and mutated phage-plasmid copies. In that scenario there may be some escape in the repression which could lead to an increase in the PPCN (and maybe even to lytic events?). However, the authors also showed quite convincingly (figures s8 and s9), that those wild-type homophage-plasmid cells that recovered this genotype due to segregational drift after being heterozygous, presented an increased PPCN compared to the ancestral ones. I don't really understand why that would be the case. If the wild-type repressor has recovered the original levels the PPCN should be mostly controlled by the plasmid replication initiation protein and, unless there are other mutations in the plasmid region (or the chromosome) affecting PPCN, I would expect the PPCN going back to the original one. Do the authors have an explanation for this observation?

Minor comments:

- line 60: infected and "resistant" cells

-Figure S6 is difficult to understand, aren't the genes supposed to be labeled?

-line 211, 40-50K?

Alvaro San Millan

Reviewer #1 (Remarks to the Author):

This very interesting manuscript describes a study on the evolution of lysis-lysogeny decision of phage-plasmids in an environmental bacteria. Phage-plasmids have recently spurred significant interest since they are more abundant than previously thought and have genetic traits of both phages (horizontal transmission) and plasmids (vertical transmission). Here it is shown that a phage-plasmid evolves to become more lytic and that it can super-infect hosts carrying the original phage-plasmid. This leads to heterozygosity in phage-plasmids which affect the fate of the cell. This is very interesting, because there are quite a few works on heterozygosity in plasmids but little or nothing in phages. The manuscript is clearly written and the figures illustrate well the results. Supplementary material is useful and almost complete (see below). Overall, I found this to be a very interesting and timely contribution.

Thank you for the positive comments. We find the comments below very helpful for improving our manuscript. Please find the point-to-point response detailed below.

-I was expecting some longer discussion on the observed evolution towards higher virulence in the context of the experience and how likely it is of being important in natural populations. I expect that the environment of the experiments selects for it, right? (phages have only lysogens to infect). How relevant is this outcome expected to be in nature?

Indeed, the serial dilution regime in our experiment selects for faster proliferation, not only for bacteria but also for phage-plasmids. Consequently, higher virulence tends to be selected for as they generate much more offspring through lytic infections, compared to the temperate counterparts. In natural populations, this can be also the case when the host, *Tritonibacter mobilis* as a copiotrophic marine heterotroph, grow rapidly in nutritional hotspots such as phycosphere. Under that scenario, the virulent phage-plasmids might gain an evolutionary benefit in producing more descendants, but without immediately wiping off all the hosts. Then the released virulent phage-plasmid particles can continue to re-infect other lysogenized hosts or hunt for new susceptible hosts. We have revised the discussion in the main text to address this. Please refer to L245-L250 for changes.

-"we conclude that phage plasmids are evolutionary chimeras that combine elements with disparate evolutionary histories and disseminate across vast geographic distances " This general claim seems unwarranted given that the authors have analyzed only three elements. Furthermore, this analysis does not allow to orient the events, i.e. to determine which gave rise to what. This is important because plasmids and phages also recombine a lot. So, how these elements came about here it's not easy to infer. I think the observation of homology between phage-plasmid, phages and plasmids is interesting and worthy of note. The speculation that the phage-plasmid arose as a chimera has merit. But given the limited evidence, I don't think we can conclude from these results that this phage-plasmids (and even less these elements in general) originate from a chimera.

We agree with you that more evidence will be needed to orient the evolutionary event mentioned above. In light of this, we have made corresponding changes in the main text to avoid any strong statement of the evolutionary origin of phage-plasmids when we describe what we found through genomic data analysis. Please refer to L255-L256 for revision.

-L101. I was expecting some discussion around the fact that many of the mutations are slippage of tandem repeats. These mutations are often reversible, which could be relevant as a way of regulating the lysis-lysogeny switch (Moxon's, Current Biol, classical paper on contingency loci might be more relevant than the papers about mutations in autism). This mechanism might provide an interesting reversible mutational switch and sustain the points of discussion around lines 244.

We appreciate this insightful comment. In addition to Moxon's review, we found a few more interesting studies after that (and potentially inspired by that) reporting reversible phenotypic switches enabled by expansion/shrinkage of tandem repeats. We realize that this is an interesting point very relevant to our findings. Please refer to L102-L104 for revision.

-There are a number of recent works showing that super-infection exclusion is not as ubiquitous as previously thought, e.g. Bondy-Denomy, ISMEj, 16, Sousa, ISMEj, 20, etc. Also, there is theoretical work on the short and long-term advantages of exclusion in Hunter, PLoS CB, 22 and Wodarz, Evol Lett, 18. It would be interesting (and might increase the paper readership) to frame the discussion in this broader literature (even if it concerns other temperate phages).

This is another insightful comment. Indeed, our observation of re-infection echoes these recent studies on the point that super-infection exclusion is not as ubiquitous as what we previously imagine based on the textbook cases of phage lambda. Accordingly, we have revised our discussion in this context. Please refer to L227-L229 for revision.

-The section on the simulations should include a description of implementation and the code should be made available.

We have added a description of implementation in the Method part, in which we also include the Github link to the R script we wrote for simulation. Please refer to L406-L409 for revision.

Minor points.

-L76. At this stage in the text it is not clear enough to the reader how productive infection was observed. In these studies one often delivers phages to observe productive infection. Not the case here, if I understood correctly. So, this should be more clearly explained from the onset. Maybe the issue about super-infection being tolerated in this system should be brought forward earlier in the text.

Here, the observation of productive infection was based on 1) fold increase of phage-plasmid genome multiplicity as calculated from sequencing (Fig. 1A and Fig. S4) and 2) observation of free virion particles under electron microscope. We have revised L76-L85 in the manuscript to avoid potential confusion.

-Figure 1D should include other genetic elements in the annotation. It indicates one gene, but not the others, some of which might have mutations too (since some mutations are outside of the gene).

In fact, there is only one gene identified within this ~1kbp region where we observe mutations. As shown in Figure 1D, although most mutations are either in the HTH domain or the putative promoter region upstream, there are four mutations occurring in non-coding region. Intriguingly, this non-coding region

is also transcribed, as indicated from the level of transcripts in Figure 1D. Therefore, we speculate this non-coding region might play a role in post-transcription processing. However, the effort to test this hypothesis can be complicated as this environmental bacterium tends out to be genetically intractable although we have tried several methods of genetic manipulation.

-Figure S6 would benefit if the most outstanding transcripts were labelled with gene names. This is a very good suggestion. Please refer to the updated Figure S6 with labels added.

-The authors mention the availability of expression and genomic data upon publication. I think that one supplementary table is missing. It would have the genes of the phage-plasmid in lines, and in columns the information on their annotation (position, protein ID, and predicted function), information on those that have homologs in Fig4 and the levels of expression in the two conditions (fig S6). Otherwise it's difficult or impossible to access that information.

We have added a new Table S2 to the supplementary material regarding the details of phage-plasmid genes, including 1) protein sequences, 2) protein position, 3) annotated function, 4) expression in the two conditions, 5) homolog information in Figure 4. The official ID for each protein has not been assigned by NCBI yet as they are still processing the genome files. Therefore, we have directly provided in this table the protein sequences in order to facilitate any potential uses by readers. It will be very easy to link back to NCBI IDs once they are officially assigned and made public.

There are a few typos or issues with style:

- L32. Escherichia coli P11 infecting Escherichia coli
- L247. plasmids such the one here
- L356. used to search for search
- L376. switches to be lytic due to silence of the phage repressor

Thank you so much for pointing out these typos. We have carefully revised the manuscript. Please refer to L31, L251, L360 and L379-L380 in the revised manuscript.

A few issues with references (see also above):

- L436. "Recent studies suggested that specificity of phage infection may be related to the structure of bacterial capsule " the references missing

Sorry for missing the references here! Please refer to L444-L445 for revision.

- Some references have twice the year of publication: 6, 7, 8, etc
- Some preprints have been published or have different format from the rest: 5, 22, etc
- Some references have links in titles or other oddities: 11, 17, etc

We apologize for these format errors and thank you for pointing them out! It seems there is some issue with the software we use for formatting the references. We have carefully examined all the references and performed the following corrections: 1) deleting the duplication of publication years, wherever applicable 2) updating Ref.5 which has now been published on *mBio* and Ref. 44 which has now been published on *Curr. Biol.* After searching the internet we think Ref. 25 is still on *bioRxiv*, so we keep it

there and revised the format, 3) removing website links, if any.

Reviewer #2 (Remarks to the Author):

In this study, Shan, Szabo and Cordero investigate the evolutionary dynamics of a phage-plasmid in the marine bacterium *Tritonibacter mobilis*. The authors describe how the phage-plasmid became lytic after 35-40 generations of propagation of bacterial populations in the lab due to mutations in the phage repressor. However, after an extremely rapid increase in frequency, the mutant phage alleles did not reach fixation but stabilized at around 60% frequency in the population. These results suggested horizontal transfer of the derepressed phage-plasmid, producing heteroplasmy in reinfected cells carrying wild type phage-plasmids. The authors confirmed this possibility experimentally, and demonstrated how reinfection and segregational drift could explain the evolutionary dynamics observed. Finally, they also showed that similar phage-plasmids to the one described in their study (or phage-plasmid combinations carrying a very similar plasmid or phage section) are common and distributed around the world, suggesting a modular mode of evolution of phage-plasmid combinations, and the generalized relevance of the eco-evolutionary dynamics described in this study. In general, I think this is a very interesting work, providing new insights into phage-plasmid biology and their impact in bacterial evolution. I'm a big fan of the role of multicopy plasmids in bacterial evolution, and this study provides yet another fascinating example of the relevance of copy number of extrachromosomal genetic elements determining evolutionary dynamics in bacteria. In summary, I have enjoyed reviewing this paper a lot and I think it will represent an important contribution to the field. There is only one aspect of the paper that I don't fully understand (see comment below).

Thank you very much for your positive comments on our work. Importantly, we have benefited a lot from previous studies on the evolution of multicopy plasmids when we carried out our study. The insights from those previous studies, such as heteroplasmy, segregational drift, etc. has laid a very strong foundation for us to better understand the phage-plasmid in this study.

Major comment

I have to start by stating that I am an expert on plasmid biology, so some of the aspects of phage biology may be beyond my knowledge.

In the wild-type cells the copy number of phage-plasmid per chromosome is close to one. Once the mutations in the repressor appear, the phage-plasmid copy number (PPCN) goes up, which is probably mostly due to the active lytic cycle in some of the cells. However, it is also possible that in hetrophage-plasmid cells (cells with both wild-type and mutated phage-plasmids) the PPCN can be increased due to the dilution of the wild-type repressor, which is only produced by the wild-type phage-plasmid, but has to bind both the wild-type and mutated phage-plasmid copies. In that scenario there may be some escape in the repression which could lead to an increase in the PPCN (and maybe even to lytic events?). However, the authors also showed quite convincingly (figures s8 and s9), that those wild-type homophage-plasmid cells that recovered this genotype due to segregational drift after being

heterozygous, presented an increased PPCN compared to the ancestral ones. I don't really understand why that would be the case. If the wild-type repressor has recovered the original levels the PPCN should be mostly controlled by the plasmid replication initiation protein and, unless there are other mutations in the plasmid region (or the chromosome) affecting PPCN, I would expect the PPCN going back to the original one. Do the authors have an explanation for this observation?

Thank you for your insightful question. As you pointed out, increased PPCN of homo-phage-plasmid cells supports our hypothesis of segregational drift. We also agree with you that the wild-type PPCN in these cells should, in theory, finally go back to one again, given sufficiently long time unless new mutations arise. In fact, this is pretty consistent with our observation - in Figure S8, those post-segregational drift populations with only wild-type phage-plasmids have average PPCN ranging from 1.4 to 1.8, suggesting that some of the hosts are likely to already have only one copy of phage-plasmid, while others still have more than one copies. Therefore, on the whole population level it might undergo a process of losing excessive copies of wild-type phage-plasmids, which would take some more generations to finally reach PPCN near or equal to one. We hope this explanation could address your question.

In addition, although we have confirmed the occurrence of segregational drift and showed that it is *sufficient* to explain the observed evolutionary patterns, there can be still other factors also playing a role in the productive infection. For instance, we agree with you that the dilution of wild-type within a host cell might be a possibility, e.g., a host cell with one copy of wild-type phage-plasmid and four copies of mutated phage-plasmid might have a higher probability of suffering from a productive switch.

Minor comments:

-line 60: infected and "resistant" cells

Sorry for the typo. Please refer to L60 for revision.

-Figure S6 is difficult to understand, aren't the genes supposed to be labeled?

Thank you for this suggestion! We have updated Figure S6 to add labels there.

-line 211, 40-50K?

Yes, indeed. We have corrected this typo in L211.